# DARA: A Dynamic Augmented Reality Architecture for Human-Robot Interaction

Brennan Miller-Klugman
brennan.miller_klugman@tufts.edu
Tufts University
Medford, Massachusetts, USA

Andre Cleaver
andre.cleaver@tufts.edu
Tufts University
Medford, Massachusetts, USA

Jivko Sinapov
jivko.sinapov@tufts.edu
Tufts University
Medford, Massachusetts, USA

## ABSTRACT

In recent years, Augmented reality (AR) has been increasingly studied as a medium to facilitate human-robot interactions (HRI). This paper presents DARA (Dynamic Augmented Reality Architecture), a novel mobile-based AR architecture that enables simultaneous connections to multiple distinct robots. Utilizing a database to manage configurations, DARA creates connections to robots and instantiates visualizations, called augments, at runtime allowing new robots to be added without requiring the application to be rebuilt.

## KEYWORDS

Augmented-Reality, ROS, HRI

**ACM Reference Format:**
Brennan Miller-Klugman, Andre Cleaver, and Jivko Sinapov. 2023. DARA: A Dynamic Augmented Reality Architecture for Human-Robot Interaction . In *Proceedings of VAM-HRI '23: International Workshop on Virtual, Augmented, and Mixed-Reality for Human-Robot Interactions, (VAM-HRI '23)*. ACM, New York, NY, USA, 3 pages. https://doi.org/10.1145/nnnnnnn.nnnnnnn

## 1 INTRODUCTION

Robotics platforms collect and create a myriad of data ranging from camera feeds to control signals. Figuring out ways to visualize these large quantities of data that are collected in real-time is critical to help humans to understand the internal states of robots. Augmented Reality (AR) provides a unique medium of data visualization, called augments, wherein pertinent information can be overlaid on top of the physical world. These augments can take on a variety of different shapes and sizes ranging from displaying motion intent [7] to creating augmented digital twins [8].

In this paper, we introduce DARA (Dynamic Augmented Reality Framework), a novel AR framework that features the following contributions:

(1) Provides a system to further abstract augmented visualizations and allow them to be recycled and applied to new and distinct robot platforms via configuration files.
(2) Supports connections to multiple robots simultaneously.

(3) Uses a database to store the connection information needed to communicate with the robot, as well as augment configurations, that are used to instantiate visualizations at runtime.

As shown in the high-level systems diagram in Fig. 1, at runtime DARA queries the database and uses the contained information to open a TCP connection and to dynamically instantiate prefabricated augment objects, created by the end-user, for each robot. DARA also includes a web interface that allows users to add new robotics platforms or update existing configurations, without needing to modify any code. The ability to connect to and view data streams from multiple robots simultaneously enables further exploration into using AR in a fleet setting.

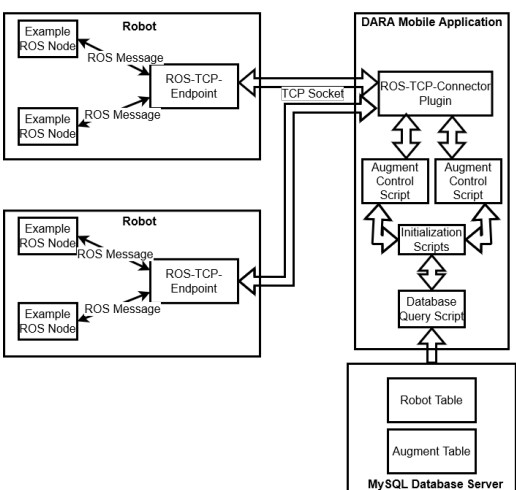

**Figure 1: DARA System Diagram With Two Connected Robots**

## 2 RELATED WORK

In recent years AR has been increasingly explored as a medium to facilitate HRI. Researchers have used AR to teleoperate robots [2]. Mott et al. in [3] presents the use of AR to facilitate communication from a robot to a human working as a team in a search and rescue environment. Other work has looked at using AR to display virtual representations of robots [6, 7].

Given that different robots possess different sensors and actuators, augments are often tailor-made for specific platforms and applications. Some, more generalized frameworks have been proposed that allow for multiple distinct types of visualizations in a single platform [1, 4]. Cleaver et al. developed an augmented reality application capable of displaying visualizations like lidar

point clouds, and battery status in [1]. These systems, while more robust than platforms made for a single application, do not offer a streamlined process for adding new robots and creating new augments.

Our goal is to expand on AR for HRI by creating a more general AR architecture that simplifies the process of adding new robots and augments. By parsing configurations at runtime DARA's augments are recyclable and can be applied to new robots by changing parameters in the database.

## 3 METHODOLOGY

DARA[1] was built in the Unity[2] game engine and leverages Vuforia[3]. Robots that interface with DARA run the Robot Operating System (ROS) [5] and communicate with the Unity application through the ROS-TCP-Connector[4] plugin which opens a connection to a ROS-TCP-Endpoint[5] running on the robot. A general system diagram of DARA with two connected robots can be seen in Fig. 1.

Unity requires applications to be rebuilt after even the smallest changes. These builds can take a considerable amount of time, which scales depending on how large the codebase is. By allowing configurations to be adjusted at runtime, new robots can be added to the system without requiring the Unity project to be rebuilt. DARA implements a MySQL[6] database to store configuration information that is queried by the application at runtime. DARA's database consists of two tables, a robot table, and an augment table.

### 3.1 Robot Table

The robot table, shown in Fig. 2 contains the information necessary to open a TCP connection to the robot. This table includes a unique identifier, a name for the robot, and the IP and port of the ROS-TCP-Endpoint running on the robot. Additionally, in the database each robot is assigned a unique image tag which is used by the mobile application to sync with the robot.

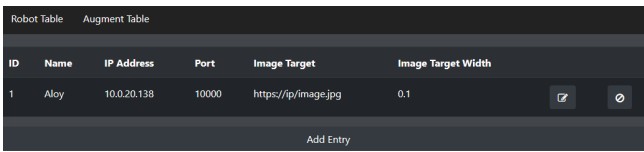

Figure 2: Robots database table shown in DARA's web interface

### 3.2 Augment Table

The augment table, shown in Fig. 3 contains the information required by the mobile application to render augments. This table consists of a robot id, which is the id number of the robot the augment is associated with, an augment name, the location of the augments Unity prefab, and a link to a configuration file. The configuration file is a JSON file that contains information relative to an augment.

---

[1]DARA's source code can be found at https://github.com/brennanmk/DARA
[2]https://unity.com/
[3]https://www.ptc.com/en/products/vuforia
[4]https://github.com/Unity-Technologies/ROS-TCP-Connector
[5]https://github.com/Unity-Technologies/ROS-TCP-Endpoint
[6]https://www.mysql.com/

When the augment prefab is instantiated, the configuration file is parsed for the information needed to use the augment.

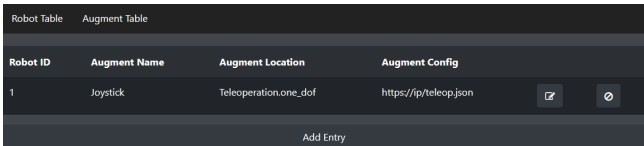

Figure 3: Augments database table shown in DARA's web interface

DARA displays augments using an Android-based application that was tested on both tablets and phones. When the app is launched, the user is prompted to scan a robot tag. When the tag is recognized by Vuforia, the application attempts to connect to the ROS-TCP-Endpoint of the robot. If the connection is successful, the user can either select to scan additional robot tags, or continue to the augment interface. If the connection fails, the user is prompted to retry.

Upon launching the augment interface, the augment database table is queried and the Unity prefabs for each item are instantiated. Additionally, a menu that allows augments to be toggled is created dynamically and is pictured in Fig. 4.

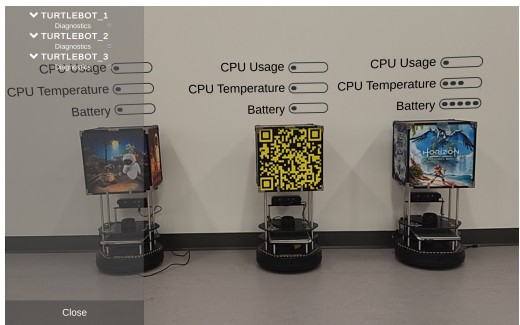

Figure 4: DARA's mobile application with three connected robots. The application menu is shown on the left and allows users to select which augments to enable. Above the three Turtlebot2's, diagnostic information is displayed as an augment.

Built using Flask[7], a Python based web framework, DARA's web interface allows users to add new robots and edit configurations of existing robots. The interface is split into two parts, the robot table, shown in Fig. 2, and the augment table, pictured in Fig. 3.

## 4 CONCLUSION

This paper presented DARA, a novel mobile-based AR architecture that enables simultaneous connections to multiple distinct robots. By utilizing a database to store information related to connecting to robots and loading augments at runtime, DARA is able to support new platforms without needing to be rebuilt and redeployed. Using

---

[7]https://flask.palletsprojects.com/en/2.2.x/

a custom web-based application, modifying database contents can be done without needing to construct SQL code. In the future, we hope to continue work on DARA by creating new augments and expanding to support more platforms (like IOS). We plan to utilize DARA to conduct HRI studies in order to explore how AR can be used to enable interactions with multiple robots simultaneously.

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
