# OpenReview forum: "DARA: A Dynamic Augmented Reality Architecture for Human-Robot Interaction"
_humanrobotinteraction.org/HRI/2023/Workshop/VAM-HRI — VAM-HRI 2023 Oral_

### Official Review · Program_Chairs · 2023-02-25
**Accept**

**Rating:** 7
**Confidence:** 5

**Review:**

Reviewer 1:

This paper presents DARA, an architecture for facilitating HRI with multiple robots and AR devices. Overall I recommend this paper be accepted, it is relevant to the community and addresses an important problem in multi-agent HRI with VAM technologies.

My main feedback is that I suggest the authors propose a way to evaluate and measure how their system helps improve VAM-HRI. It seems the biggest benefit comes from not needing to rebuild applications (which is costly in time), so an experiment where users are developing VAM-HRI systems with DARA vs. traditional development and analyzing things like time to complete, user experience, etc. would be beneficial to understand how useful the system is.

Reviewer 2:

This paper presents a novel architecture for connecting multiple robots and displaying their “augments” in a single display. This design could be the basis for some very useful developments in the VAM-HRI community. Some suggestions for improving the paper follow:

Please make sure you are using the correct template (ACM).

Figures 2 and 3 are very difficult to see. If you want to include them, consider making them larger (perhaps banner type) so that the text is readable.

The authors use the term “mobile-based” and “Android,” however it would be helpful for readers to know specifically what kind of hardware this was deployed on.

The paper mentions that there are options for “Augments” that the user can choose, however there is no list provided of what those Augments might be. Can the authors clarify whether these are pre-fabricated options that come with DARA or if they must be designed by the user?

---

### Decision · Program_Chairs · 2023-03-02

Accept (Oral)